# Exploiting endogenous fibrocartilage stem cells to regenerate cartilage and repair joint injury

Mildred C. Embree[1], Mo Chen[2], Serhiy Pylawka[1], Danielle Kong[2], George M. Iwaoka[2], Ivo Kalajzic[3], Hai Yao[4], Chancheng Shi[5], Dongming Sun[6], Tzong-Jen Sheu[7], David A. Koslovsky[8], Alia Koch[9] & Jeremy J. Mao[2]

Tissue regeneration using stem cell-based transplantation faces many hurdles. Alternatively, therapeutically exploiting endogenous stem cells to regenerate injured or diseased tissue may circumvent these challenges. Here we show resident fibrocartilage stem cells (FCSCs) can be used to regenerate and repair cartilage. We identify FCSCs residing within the superficial zone niche in the temporomandibular joint (TMJ) condyle. A single FCSC spontaneously generates a cartilage anlage, remodels into bone and organizes a haematopoietic microenvironment. Wnt signals deplete the reservoir of FCSCs and cause cartilage degeneration. We also show that intra-articular treatment with the Wnt inhibitor sclerostin sustains the FCSC pool and regenerates cartilage in a TMJ injury model. We demonstrate the promise of exploiting resident FCSCs as a regenerative therapeutic strategy to substitute cell transplantation that could be beneficial for patients suffering from fibrocartilage injury and disease. These data prompt the examination of utilizing this strategy for other musculoskeletal tissues.

[1] TMJ Biology and Regenerative Medicine Laboratory, College of Dental Medicine, Columbia University Medical Center, 630 W 168th St, P&S 16-440, New York, New York 10032, USA. [2] Center for Craniofacial Regeneration, College of Dental Medicine, Columbia University Medical Center, 622 W 168th St, New York, New York 10032, USA. [3] Department of Reconstructive Sciences, MC3705, L7005, University of Connecticut Health Sciences Center, 263 Farmington Avenue, Farmington, Connecticut 06032, USA. [4] Clemson-MUSC Bioengineering Program, Department of Bioengineering, Clemson University, 173 Ashley Avenue, MSC 508, Charleston, South Carolina 29425, USA. [5] Chongqing Institute of Green and Intelligent Technology, Chinese Academy of Sciences, 266 Fangzheng Avenue, Shuitu Hi-tech Industrial Park, Beibei District, Chongqing 400714, China. [6] W.M. Keck Center for Collaborative Neuroscience, Rutgers, The State University of New Jersey, 604 Allison Road, D-251, Piscataway, New Jersey 08854, USA. [7] Center for Musculoskeletal Research, University of Rochester Medical Center, 601 Elmwood Avenue, Box 665, Rochester, New York 14620, USA. [8] Metropolitan Oral Surgery Associates, 488 Madison Avenue, #200, New York, New York 10022, USA. [9] College of Dental Medicine, Division of Oral and Maxillofacial Surgery, Columbia University Medical Center, 622 W 168th St, New York, New York 10032, USA. Correspondence and requests for materials should be addressed to M.C.E. (email: mce2123@cumc.columbia.edu).

Fibrocartilage tissues include the knee meniscus, tendon-bone junction, intervertebral disc and temporomandibular joint (TMJ) condylar cartilage (CC) and disc[1]. Unlike hyaline cartilage, fibrocartilage consists of various proportions of both fibrous and cartilaginous tissue pending tissue-specific biomechanical functional demands and provides both tensile and compressive strength, respectively[2–4]. Given the restricted number of cells and lack of vascular supply, cartilage has relatively poor regenerative properties[5]. Thus fibrocartilage trauma and diseases, including knee meniscus injury[6] and TMJ[7] or intervertebral disc[8] degenerative disease, can cause permanent tissue loss and disability. Clinical treatment modalities for fibrocartilage trauma or disease are limited and involve either palliative care[9] or invasive surgical interventions that often fail or cause further tissue damage[10,11]. Minimally invasive cell-based therapies that prevent fibrocartilage degeneration or promote repair are not available clinically.

Increasing efforts have been made to develop stem cell-based therapies for musculoskeletal tissue regeneration that involve stem cell expansion and transplantation[12,13]. However, the success of this approach is dependent upon overcoming multiple obstacles, including immune rejection, pathogen transmission, potential tumorigenesis and host tissue engraftment[5,14,15]. An alternative approach involves tissue regeneration achieved by the recruitment and stimulation of stem/progenitor cells[16–18]. Hence, therapeutic strategies that harness resident stem cells[19] to repair and maintain adult musculoskeletal tissue homoeostasis could be a minimally invasive cell-based treatment option that circumvents critical barriers to stem cell transplantation.

The TMJ condyle is a fibrocartilage tissue with well-organized cellular zones of maturation. These zones include a distinct fibrous superficial zone (SZ) followed by proliferative/polymorphic, chondrocyte and hypertrophic chondrocyte zones of maturation[20–22]. Although the cellular origin of the TMJ is unclear, studies suggest that neural crest cells and mandibular bone periosteum cells contribute to TMJ formation[23]. The TMJ cellular condensation forms a cartilage anlage that is resorbed by osteoclasts and is replaced by mineralized bone[21]. Unlike long bone growth plate, at skeletal maturation the TMJ cartilage is not completely resorbed and is maintained like an articular joint cartilage[20]. In adults, the TMJ condyle is lined with a fibrous, SZ tissue enriched in type I collagen and lubricin, while the deeper cell layers are comprised of cartilaginous matrix consisting of type II collagen and aggrecan[22,24]. Although the precise function of the fibrous SZ tissue is unknown, we speculate that the SZ tissue stores a reservoir of stem cells that give rise to mature chondrocytes and osteoblasts critical for TMJ development and homoeostasis[25]. Moreover, the cell signalling pathways regulating cell fate determination during TMJ development is unclear. Given that inhibited canonical Wnt signalling is critical for skeletal stem cell fate specification toward cartilage lineage[26–28], we surmise that Wnt signals may be involved in TMJ stem cell fate specification toward chondrocytes. Furthermore, Wnt signals are also critical for cartilage homoeostasis, and induce chondrocyte terminal differentiation[29–32].

Here we show for the first time that the fibrous SZ tissue in the TMJ condyle is indeed a niche that harbours fibrocartilage stem cells (FCSCs). We discover a single FCSC is capable of not only generating cartilage and bone but also organizing a haematopoietic microenvironment (HME)[33,34] when transplanted in vivo. To harness the cartilage regenerative capability of FCSCs, we show that suppression of canonical Wnt signals promote FCSCs to differentiate into chondrocytes. Our results further reveal that overactive Wnt signals disrupt fibrocartilage homoeostasis and cause degeneration by depleting the FCSC pool. We take advantage of endogenous FCSCs and present a scaffold free, non-surgical therapeutic approach to exploit their ability to regenerate cartilage and repair joint injury. We discover that therapeutic application of exogenous canonical Wnt inhibitor sclerostin (SOST) maintains FCSC pool and repairs cartilage using a rabbit TMJ injury model[35]. Our results reveal that careful regulation of the canonical Wnt pathway promotes FCSC to repair and regenerate injured or diseased cartilage for therapeutic applications. Our findings demonstrate for the first time a minimally invasive stem cell-based approach to repair and regenerate fibrocartilage tissue.

## Results

**The TMJ superficial zone provides a niche for FCSCs**. To localize putative, slow-cycling FCSCs, a pregnant rat was injected with EDU during TMJ morphogenesis (Fig. 1a)[36]. The percentage of EDU$^+$ label-retaining cells (LRCs) were quantified in the TMJ condyle SZ and compared with the percentage of LRCs in the remaining CC zones, including polymorphic, chondrocyte and hypertrophic zones (Fig. 1b,c). In newborns (NBs) (Fig. 1b), the percentage of SZ LRCs was significantly higher than the percentage of CC LRCs (Fig. 1c). By 8 weeks, only <0.04% CC LRCs were present (Fig. 1c), suggesting that slow-cycling cells did not preferentially reside within CC maturation zones. On the other hand, at 8 weeks nearly all LRCs were localized within the SZ (Fig. 1b, arrows). The percentage of SZ LRCs remained consistent following 16-week chase period (Fig. 1c), suggesting that putative slow-cycling stem cells may preferentially reside within SZ tissue.

To further distinguish and analyse SZ from CC tissue, SZ tissue was surgically dissected from the CC tissue in an 8-week-old Sprague Dawley rat (Fig. 1d). To confirm clean surgical separation of SZ from CC tissue, we compared their protein and gene expression profiles. Immunohistochemistry showed that lubricin[37] was expressed in the SZ tissue and aggrecan (ACAN), a marker for mature chondrocytes, was expressed in the CC tissue (Fig. 1e). ECM markers type I collagen (col1a1) and lubricin (prg4) were upregulated in SZ relative to CC tissue (Fig. 1f)[20,37]. However, cartilage markers (sox6, sox9, acan, col2a1 and col10a1) were significantly decreased in SZ compared with CC tissue (Fig. 1g). These data suggest that SZ tissue may harbour undifferentiated cells, while CC tissue may harbour mature cell phenotypes, including chondrocytes.

We hypothesized that SZ cells may give rise to mature chondrocytes localized within CC tissue. To determine the origin of mature chondrocytes in the TMJ condyle, we performed a lineage-tracing study. The skeletal stem/progenitor cell marker αSMA[38] was traced in αSMACreERT2/Ai9 transgenic mice[38] (Fig. 1h). Type II collagen expression was used to distinguish putative stem cells from mature chondrocytes. After 2 days, the percentage of αSMA$^+$col2a1$^-$ cells in SZ was significantly higher than the percentage of αSMA$^+$col2a1$^+$ cells in CC (Fig. 1i,j). By day 15, the percentage of αSMA$^+$ col2a1$^+$ cells significantly increased in CC relative to day 2 (Fig. 1i) and αSMA$^+$ cells (red) infiltrated into the mature chondrocyte col2a1$^+$ cell population (green) (Fig. 1j). In untreated αSMACreERT2/Ai9$^+$ and treated αSMACreERT2/Ai9$^-$ mice, αSMA$^+$ cells were not detected (Fig. 1j). These data provided evidence that undifferentiated αSMA$^+$ cells within SZ tissue give rise to mature, col2a1$^+$ chondrocyte progeny localized in CC tissue. Taken together, these data point to the possibility that the SZ may provide a niche for putative FCSCs.

**FCSCs are distinct from differentiated chondrocytes**. To confirm the presence of putative FCSCs within the SZ niche, single-cell suspensions of putative FCSCs were isolated from SZ tissue and compared with donor-matched CC cells (CC cells)

and/or bone marrow stromal cells (BMSCs)[39,40]. Similar to BMSCs, FCSCs expressed cell surface markers CD90, −44, −29, −105, −146, and did not express leukocyte markers CD-45, CD79a, CD11b (Fig. 2a and Supplementary Fig. 1). FCSCs had a greater capacity to propagate relative to CCs (Fig. 2b,c). FCSCs

formed sixfold more colonies than CCs (4–5%, Fig. 2d). Unlike mature chondrocytes that typically dedifferentiate in monolayer cultures[41], FCSCs maintained cartilage phenotype and expressed early transcription factors (sox5, sox6 and sox9) at day 5 relative to day 2 and late cartilage marker acan at day 10 relative to day 2

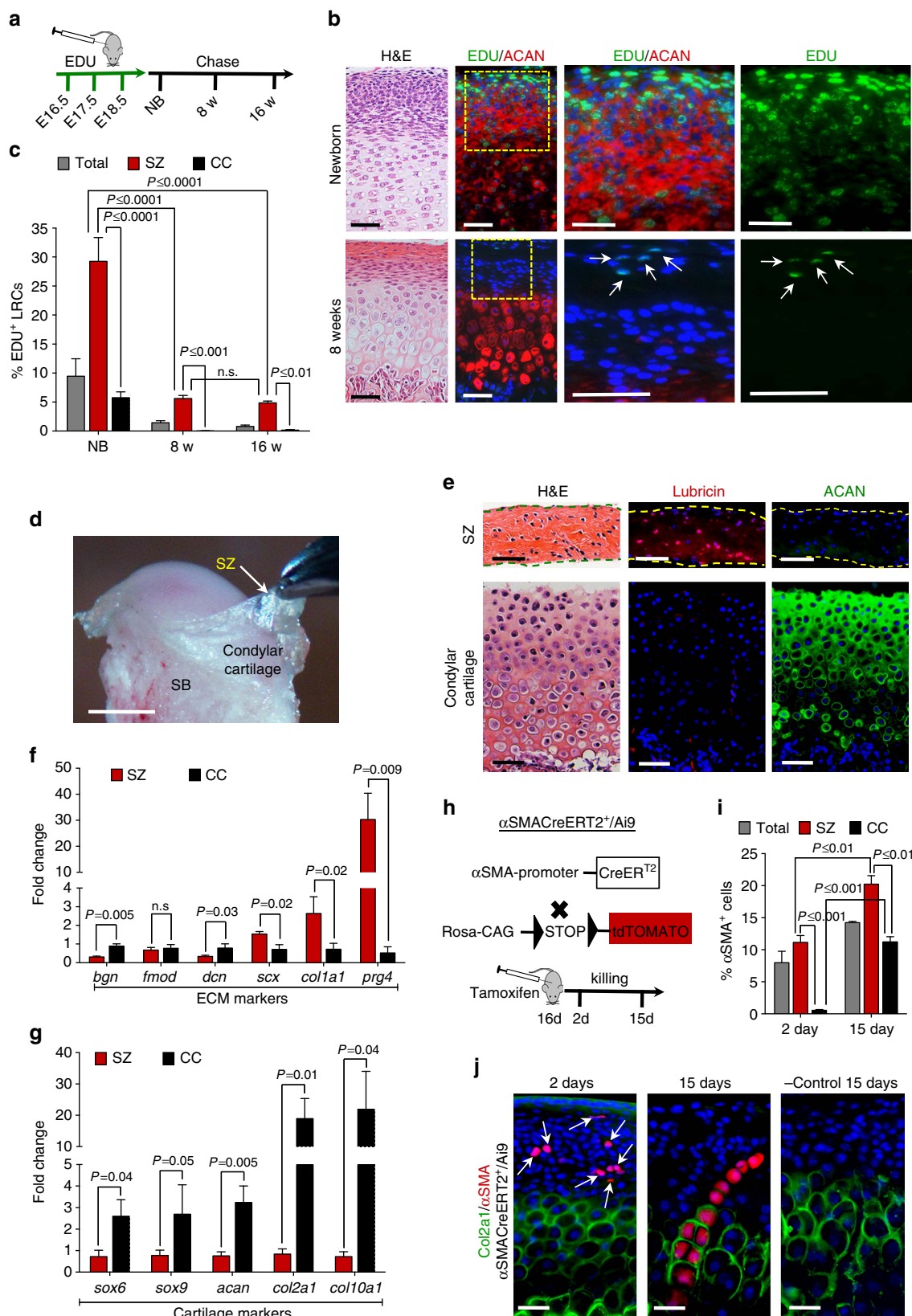

(Fig. 2e). Similar to mature chondrocytes[41], CCs were unable to sustain cartilage phenotype in monolayer cultures and demonstrated a significant decrease in *acan* and significant increase in early markers *col1a1* and *sox9* at day 10 relative to day 2 (Fig. 2f). Multipotential differentiation *in vitro* was tested in chemically defined media[40,42]. Like BMSCs, FCSCs underwent adipogenesis (oil O Red$^+$, *ppar-γ*, Fig. 2g,h), chondrogenesis in pellet cultures (Fig. 2g, ACAN$^+$) and mineralization (alizarin red$^+$, *ocn*$^+$, Fig. 2g,i). Individual colonies (31) of FCSCs showed heterogeneous differentiation potential (Supplementary Table 1; 22.5% tri-lineage, 64.5% bi-lineage and 12.9% single lineage). Thus unlike mature chondrocytes, FCSCs are capable of expansion *in vitro* and differentiation into multiple lineages.

**A single FCSC spontaneously generates cartilage and bone**. FCSCs' ability to regenerate musculoskeletal tissues was tested *in vivo* in multiple transplantation experiments (Supplementary Figs 2–4). To distinguish donor FCSCs from host cells, rat GFP$^+$ FCSCs were isolated[43] and seeded onto a collagen sponge (Supplementary Fig. 3a–c). Donor-matched BMSCs and CCs were transplanted in the same host (Supplementary Fig. 4). After 2 weeks, FCSCs (4/6 transplants) formed highly dense cellular condensations (ACAN$^+$, Supplementary Fig. 2b). By 3 weeks, transplanted GFP$^+$ FCSCs occupied the collagen sponge (Supplementary Fig. 3d) and formed cartilaginous-like tissue (5/6 transplants, toluidine blue$^+$, ACAN$^+$, Supplementary Figs 2c and 3f). At 4 weeks, transitional tissue was formed with bone, cartilage and osteoclast-mediated tissue remodelling (TRAP$^+$ purple cells, insert, Supplementary Fig. 2d) (5/6 transplants). At 8 weeks, GFP$^+$ FCSCs occupied the collagen sponge and radio-opaque, bone-like tissue formed (Supplementary Fig. 3e). GFP$^+$ FCSCs (green, white arrows, Supplementary Fig. 3g) formed trabecular bone-like tissue (OCN$^+$) with an organized HME[33] occupied by both transplanted GFP$^+$ FCSCs and GFP$^-$ host cells, including osteoblasts, adipocytes, haematopoietic cell clusters and sinusoids (Supplementary Figs 2e and 4a) (7/8 transplants). FCSCs formed bone with a HME in multiple carriers, including Matrigel (Supplementary Fig. 4a) and gel foam (not shown). Unlike FCSCs, BMSCs formed bone-like tissue in Matrigel but not collagen sponge, with no evidence of cartilage-like tissue formed (Supplementary Fig. 4b). Mature CCs did not form cartilage or bone (Supplementary Fig. 4c). A single FCSC (Fig. 3a,b) formed a single colony (Fig. 3c) and generated cartilage (Fig. 3d,e, ACAN$^+$) and trabecular bone with a HME (Fig. 3f,g, OCN$^+$) (2/17 single-cell colonies). These data demonstrate that cartilage and bone with HME[33] was generated from FCSCs. A single FCSC can differentiate into both chondrocytes and osteocytes and spontaneously recapitulate endochondral ossification *in vivo*.

**FCSCs differentiate into chondrocytes through inhibited Wnt**. We tested whether inhibited canonical Wnt signalling would direct FCSCs fate towards cartilage lineage[44,45]. Like TGF-β1 (ref. 46), treatment with Wnt antagonists[26], Dkk1 (ref. 31), Wif1 (ref. 28) and SOST[29] and small molecule Wnt inhibitor ICG-001 (ref. 47), significantly increased *acan* expression in FCSCs dose-dependently relative to vehicle control (Fig. 4a and Supplementary Fig. 5a,b). Conversely, Wnt agonist Wnt3a inhibited the expression of cartilage-related genes (*sox5*, *sox6* and *acan*) in FCSCs relative to vehicle control (Supplementary Fig. 5c). Inhibition canonical Wnt also significantly decreased FCSC proliferation over time relative to vehicle control (Supplementary Fig. 6). To directly activate Wnt signalling, FCSCs were transiently transfected with a constitutively active β-catenin mutant S33Y[48] and the Wnt luciferase reporter construct TopFlash. After 72 h, luciferase activity (Fig. 4b) and downstream Wnt target gene *ccnd1* (Fig. 4c) were significantly increased in S33y FCSCs relative to control vector, confirming Wnt signalling activation. Early cartilage-related transcription factors[46] (*sox5*, *sox6* and *sox9*) and *twist1*, a gene critical for cartilage fate specification[26], were significantly decreased in S33Y FCSCs compared with control FCSCs (Fig. 4c). Mature cartilage marker *acan* was not changed, most likely due to the early time point (Fig. 4c). These data point to the idea that inhibited canonical Wnt signalling promotes FCSC to differentiate into chondrocytes.

**FCSC pool is maintained through inhibited Wnt**. We speculated that canonical Wnt signalling was critical for regulating fibrocartilage homoeostasis[29–32]. Immunohistochemistry revealed that Wnt inhibitor SOST was ubiquitously expressed in rat TMJ condyle at 8 weeks, while β-catenin, a downstream Wnt mediator, was absent in SZ and restricted to mature chondrocytes (Supplementary Fig. 7a). Thus, immunohistochemistry suggested that canonical Wnt signalling may be temporally and spatially regulated in fibrocartilage. Consistent with these findings, gene expression of downstream Wnt targets (*ctnnb1*, *runx2*, *wisp1*, *lrp5*, *dkk1* and *wif1*) were significantly increased in condylar cartilage tissue relative to SZ (Supplementary Fig. 7b). To corroborate the role of Wnt signalling in maintaining fibrocartilage homoeostasis, SOST deficiency in mice (*sost*$^{-/-}$)[29,49] showed that *sost*$^{-/-}$ TMJ condyles had osteoarthritic changes[50], including clefting, chondrocyte clustering, and diffuse aggrecan distribution. β-Catenin expression pattern was expanded in *sost*$^{-/-}$ relative to wild-type TMJ condyles (Supplementary Fig. 8a–d), suggesting SOST deficiency enhances Wnt signalling. Degenerative changes and expansion of β-catenin expression in *sost*$^{-/-}$ TMJ condyles was also coupled with a significant decrease in the number of FCSCs localized within SZ niche relative to wild-type mice (Supplementary Fig. 8e). Similarly, β-catenin expression was present and SOST expression was absent in FCSC generated cartilage in transplantation model (Fig. 4d). Immunocytochemistry showed FCSCs had reduced β-catenin cytoplasmic accumulation in the presence of SOST after 48 h (Fig. 4e), providing further

**Figure 1 | TMJ SZ provides a niche for FCSCs.** (**a**) EDU was administered to Sprague Dawley pregnant rats ($n = 2$) daily for 3 days. EDU$^+$ LRCs were chased in newborns (NB), 8 and 16 weeks. (**b**) H&E and immunohistochemistry for aggrecan (ACAN). Arrows indicate SZ EDU$^+$ LRCs (green). Scale bar, 50 μm. (**c**) LRCs were quantified in total condyle (total), SZ, and CC using ImageJ software. Data represented are mean percentage ± s.d.; $n = 5$ Sprague Dawley rats; two-way ANOVA followed by Tukey's *post hoc*. (**d**) TMJ SZ was dissected in a 8-week Sprague Dawely male rat. SB, subchondral bone. Scale bar, 2 mm. (**e**) H&E and immunohistochemistry for lubricin and aggrecan (ACAN). Scale bar, 50 μm. (**f,g**) qRT-PCR of SZ relative to CC. Data represented are mean fold change normalized to GAPDH ± s.d.; $n = 4$ male Sprague Dawley rats 8 week; paired Student's *t*-test. (**h**) Lineage tracing experiment in αSMACreERT2/Ai9 male mice. Sixteen-day αSMACreERT2/Ai9 littermates were treated with tamoxifen. αSMA$^+$ cells quantified after 2 and 15 days. (**i**) Col2a1 expression was examined in αSMACreERT2/Ai9 mice and αSMA$^+$ cell percentages were calculated in total tissue (SZ + CC), SZ and CC. Data represented are mean percentage ± s.d.; $n = 2$ αSMACreERT2/Ai9 littermate male mice 18 and 31 days; two-way ANOVA followed by Tukey's *post hoc*. (**j**) Representative images of immunohistochemistry for col2a1 (green) and analysis of αSMA$^+$ cells (red) in αSMACreERT2/Ai9 mice treated with tamoxifen. Untreated αSMACreERT2/Ai9 littermates were negative controls. Scale bar, 25 μm. ANOVA, analysis of variance; H&E, haematoxylin and eosin; qRT-PCR, quantitative real-time PCR.

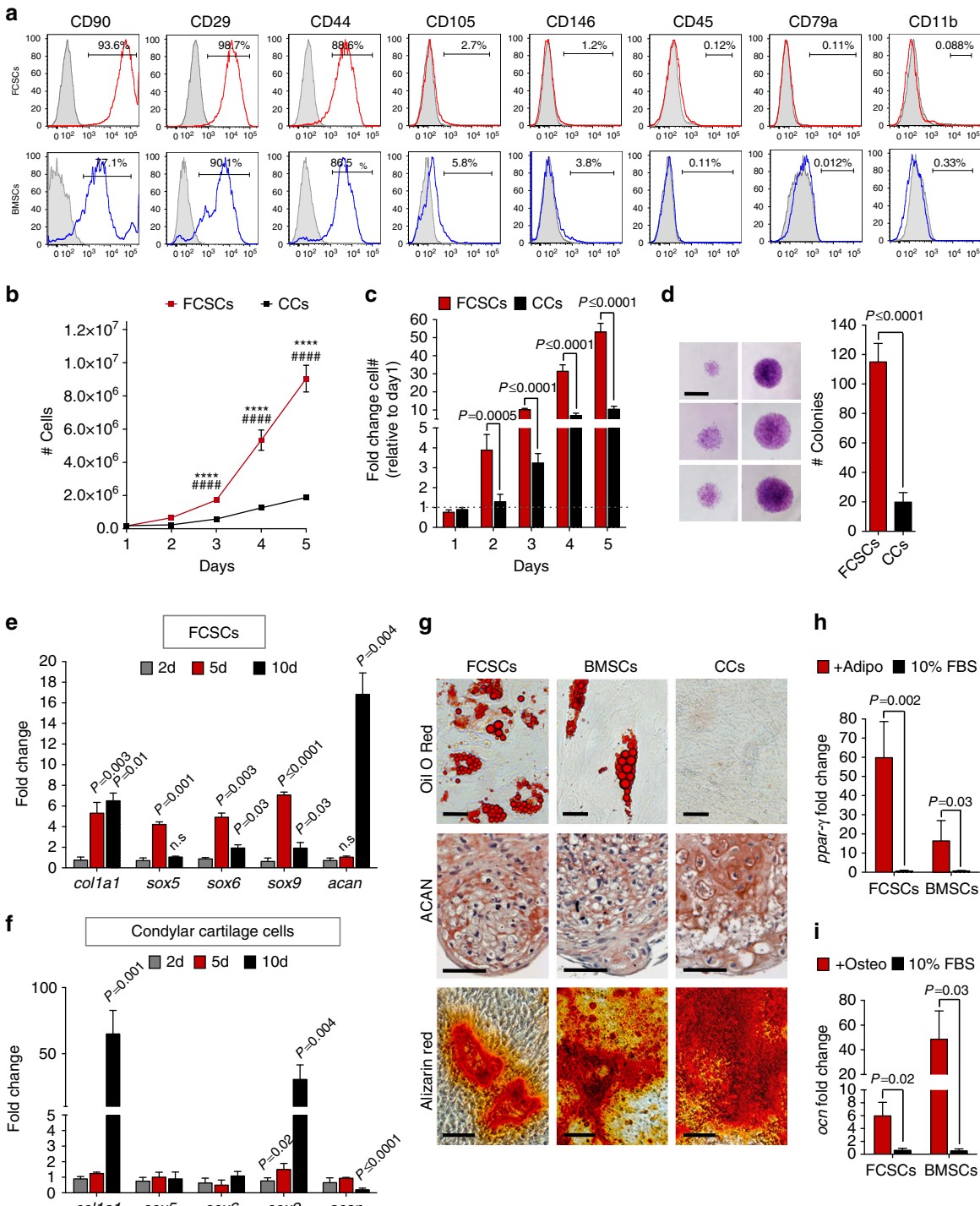

**Figure 2 | FCSCs are distinct from differentiated chondrocytes.** (**a**) Flow cytometric analysis of cell surface markers in FCSCs and donor-matched BMSCs. (**b**) Cell growth curve of FCSCs and condyle cells (CCs) number over 5 days. Data are mean ± s.d.; $n = 6$ independent experiments; ****$P \le 0.0001$ FCSCs versus CCs; ####$P \le 0.0001$ FCSCs Day1 versus FCSCs day3, 4 or 5; two-way ANOVA followed by Tukey's *post hoc* test. (**c**) Data are mean fold change in cell number relative to 1 day of FCSCs and CCs over 5d ± s.d.; $n = 6$ independent experiments; Student's *t*-test. (**d**) Colony forming assay of FCSCs and CCs. Data are mean ± s.d.; $n = 5$ independent experiments; paired Student's *t*-test. Scale bar, 2 mm. (**e,f**) qRT-PCR in FCSCs and condyle cells, respectively. Data are mean fold change normalized to GAPDH relative to 2d ± s.d.; $n = 5$ independent experiments; two-way ANOVA followed by Tukey's *post hoc* test. (**g**) Multipotential differentiation *in vitro*. Oil O red staining in lipid droplets in adipogenic media (top row, scale bar, 25 μm). Aggrecan (ACAN) immunostaining show chondrogenic potential in pellet cultures (red, middle row, scale bar, 50 μm). Alizarin red staining demonstrate calcium deposition in osteogenic media (bottom row, scale bar, 100 μm). (**h,i**) qRT-PCR of FCSCs and BMSCs in adipogenic and osteogenic media, respectively. Data are mean fold change normalized to GAPDH ± s.d.; $n = 5$ independent experiments; paired Student's *t*-test. ANOVA, analysis of variance; qRT-PCR, quantitative real-time PCR.

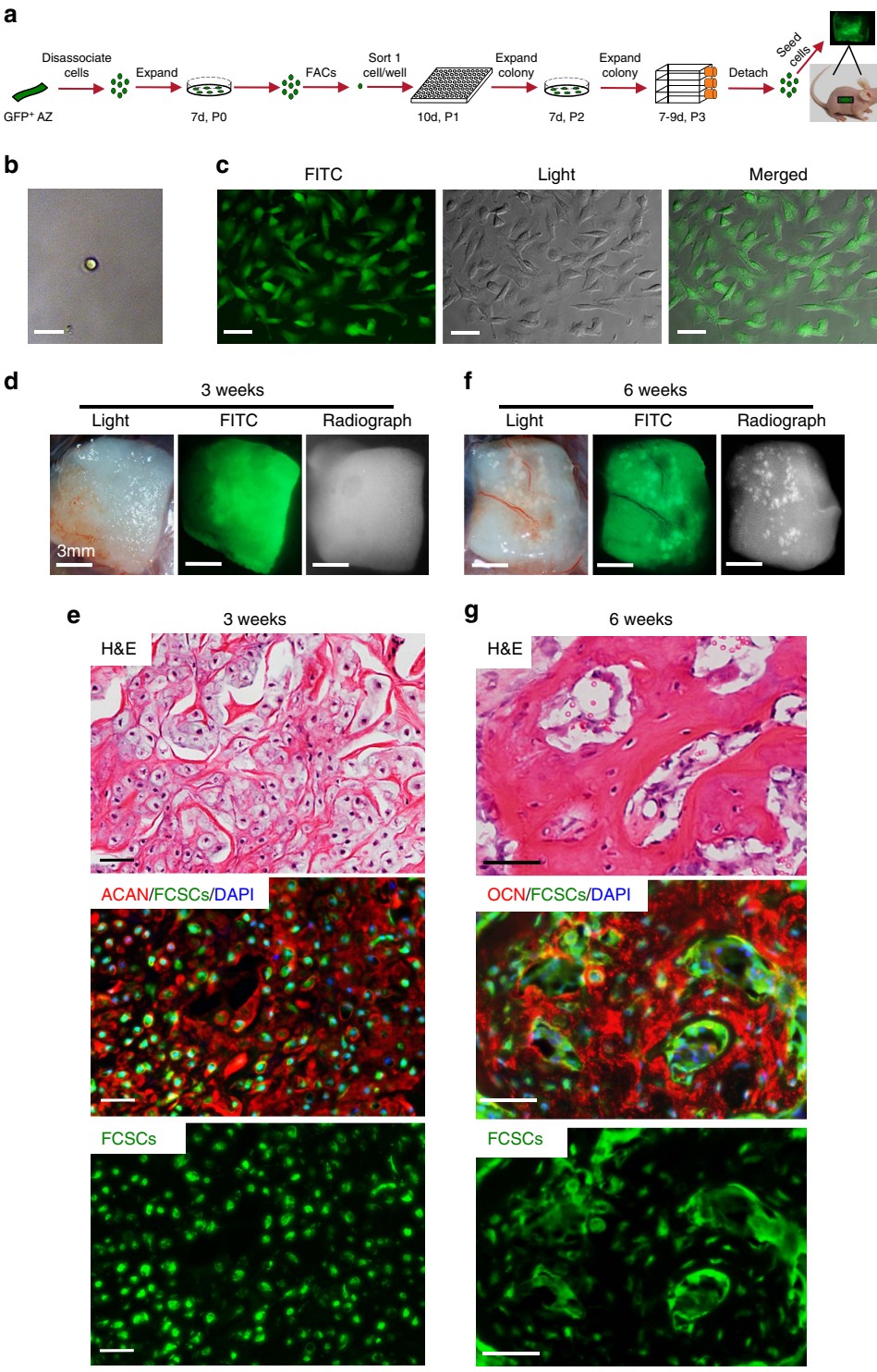

**Figure 3 | A single FCSC spontaneously generates cartilage and organized bone.** (**a**) Schematic representation showing single FCSC isolation. Heterogeneous GFP+ FCSCs were derived from the TMJ SZ tissue from male GFP transgenic rats 8 weeks. GFP+ FCSCs were expanded *in vitro* and FACs was used to plate a single-cell per well into a 96-well plate. Each single-cell colony was expanded over passages 2–3, seeded onto a collagen sponge and surgically transplanted subcutaneously on dorsum of nude mice. (**b**) Sorted single GFP+ FCSC in a single-well/96-well plate. Scale bar, 25 µm (**c**) GFP+ FCSC single-cell colony expanded at P3. Scale bar, 50 µm. (**d,f**) Collagen sponge seeded with GFP+ FCSC single-cell colony on the dorsum of nude mice after 3 and 6 weeks *in vivo*, respectively. Scale bar, 50 µm. (**e,g**) Representative H&E staining GFP+ FCSC transplant from a single-cell after 3 and 6 weeks, respectively (top row). Immunohistochemistry for aggrecan (ACAN, red) and osteocalcin (OCN, red) demonstrated single-cell GFP+ FCSC generated cartilage at 3 w and bone at 6 w, respectively (middle and bottom rows). Dapi blue staining = nuclei. H&E, haematoxylin and eosin.

support that SOST suppresses canonical Wnt signalling in FCSCs. The addition of SOST (50 ng ml⁻¹) to FCSCs for 48 h resulted in sustained cartilage formation and less bone formation relative

vehicle control (Fig. 4f,g), providing evidence that inhibited Wnt signalling maintains regenerated FCSC cartilage *in vivo*. These data point to the possibility that SOST deficiency

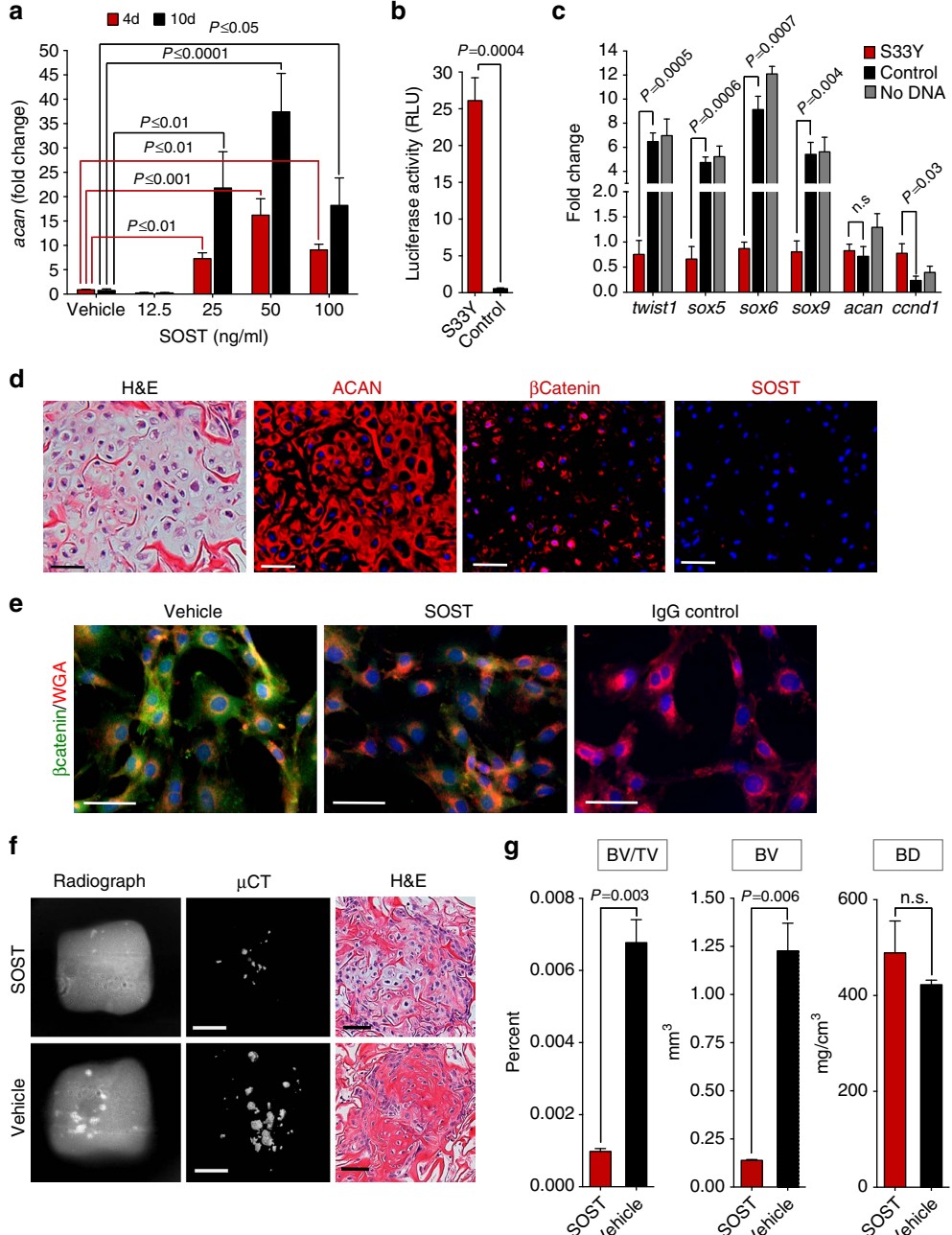

**Figure 4 | FCSCs differentiate into chondrocytes and maintain tissue homeostasis through inhibited Wnt.** (**a**) qRT-PCR show sclerostin induced aggrecan in FCSCs dose-dependently. Data are mean fold change normalized to GAPDH ± s.d.; $n = 3$ independent experiments; two-way ANOVA followed by Tukey's *post hoc*. (**b**) FCSCs were transfected with luciferase reporter TopFlash and constitutively active βCatenin$^{S33y}$ plasmid or control vector. Luciferase activity reported at 72 h. Data are mean ± s.d.; $n = 4$ independent experiments; paired Student's *t*-test. (**c**) qRT-PCR shows S33y FCSCs have decreased cartilage-related genes and increase in Wnt-related gene. Data are mean normalized to GAPDH ± s.d.; $n = 4$ independent experiments; paired Student's *t*-test. (**d**) Transplanted FCSCs showed aggrecan (ACAN) and βCatenin, but no SOST expression at 3 weeks. Scale bar, 50 μm. (**e**) Representative immunocytochemistry staining show decreased intracellular βCatenin in sclerostin treated FCSCs (50 ng ml$^{-1}$) for 48 h. Scale bar, 50 μm. (**f**) FCSCs were treated with sclerostin (50 ng ml$^{-1}$) or vehicle for 48 h and transplanted within collagen sponge in nude mice for 6 weeks. Radiographs, uCT and H&E of transplants showed cartilage was maintained and less bone formed in sclerostin treated FCSCs. White scale bar, 2 mm. Black scale bar, 50 μm. (**g**) Bone volume/tissue volume (BV/TV), bone volume (BV) and bone density (BD) in transplants seeded with FCSCs treated with sclerostin after 6 weeks. Data are mean ± s.d.; $n = 3$ transplants; paired Student's *t*-test. ANOVA, analysis of variance; H&E, haematoxylin and eosin; qRT-PCR, quantitative real-time PCR.

enhances Wnt activity and FCSC differentiation into chondrocytes, thus depleting FCSC pool.

**FCSCs regenerate and repair cartilage through inhibited Wnt.** We proposed that canonical Wnt inhibitor sclerostin maintained FCSC pool and fibrocartilage homoeostasis[29–32] and tested whether application of exogenous Wnt inhibitor would repair and regenerate injured fibrocartilage. TMJ condyle fibrocartilage degeneration and secondary osteoarthritis (OA) was induced post-traumatically in New Zealand white rabbits by creating a 2.5 mm perforation in the TMJ disc bilaterally[35]. One week

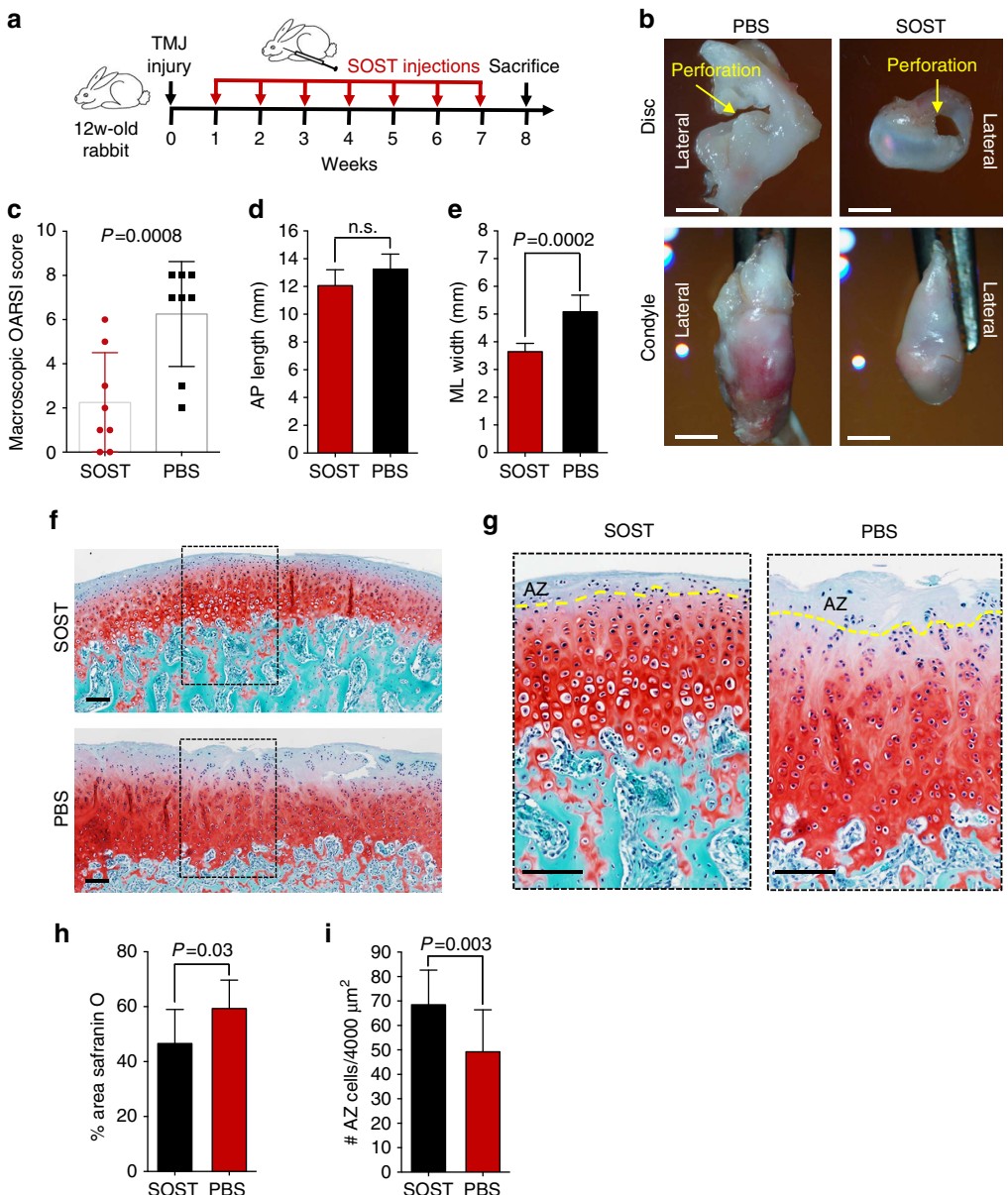

**Figure 5 | Sclerostin induces endogenous FCSCs to regenerate and repair cartilage.** (**a**) To induce fibrocartilage injury and condyle degeneration, a 2.5 mm perforation was created in 12-week rabbit TMJ discs bilaterally. One week post surgery, 0.1 ml SOST (100 ng ml$^{-1}$) and PBS was injected into contralateral sides of the inferior TMJ intra-articular space once per week for 7 weeks. (**b**) Superior view of TMJ disc and condyle 8 weeks following injury (yellow arrow) with SOST and PBS treated contralateral TMJs. Scale bar, 3 mm. (**c**) Observers ($n = 2$) blindly scored TMJ morphology using OARSI recommendations. Scatter plots with each plot representing mean rabbit score ± s.d., $n = 8$ male New Zealand White rabbits 12 weeks; paired Student's $t$-test. (**d,e**) TMJ condyle anterior to posterior length (AP) and the medial to lateral (ML) width was measured using Nis-Elements Basic Research imaging, respectively. Data are mean; error bars are s.d.; $n = 8$ male New Zealand White rabbits 12 weeks; paired Student's $t$-test. (**f,g**) Representative images of Safranin O staining of contralateral TMJ condyles treated with SOST and PBS. Yellow line = border of SZ tissue. Scale bar, 100 μm. (**h**) Safranin O staining in TMJ condyle was quantified using cellSense Imaging Software. Data are mean area ± s.d.; $n = 8$ male New Zealand White rabbits 12 weeks; paired Student's $t$-test. (**i**) The number of FCSCs quantified in SZ tissue. Data are mean ± s.d.; $n = 8$ male New Zealand White rabbits 12 weeks; paired Student's $t$-test.

following surgically induced degeneration, either 0.1 ml SOST (100 ng ml$^{-1}$) or PBS was injected into the contralateral TMJ inferior spaces once weekly for 7 weeks (Fig. 5a). SOST condyles displayed mild, surface irregularities, while contralateral PBS condyles revealed widespread, severe surface irregularities (Fig. 5b and Supplementary Fig. 9) and had significantly higher Osteoarthritis Research Society International (OARSI) recommended macroscopic scores[51] (Fig. 5c). There were no differences in condyle lengths (Fig. 5d) but the width of SOST condyles was significantly smaller than PBS condyles (Fig. 5e),

suggesting that SOST also reduced joint swelling. The area of Safranin O staining in PBS condyles was diffuse and significantly greater compared with SOST condyles, which was stained uniformly (Fig. 5f,h). The number of FCSCs within the SZ tissue was significantly reduced in PBS condyles compared with SOST condyles (Fig. 5g). SOST treatment improved TMJ gross morphology and proteoglycan distribution, prevented the depletion of FCSCs in SZ tissue and repaired cartilage in this injury model. These data reveal the possibility that manipulation of endogenous stem cells in fibrocartilage as a potential

non-invasive, stem cell-based strategy to treat fibrocartilage degeneration.

**Therapeutic strategy to exploit endogenous FCSCs in humans.** We investigated whether this minimally invasive stem cell-based therapeutic strategy could be translated to diseased fibrocartilage in humans. In 12 patients that underwent TMJ replacement surgery[52], we assessed the osteoarthritic histopathological score[53], proteoglycan content and β-catenin expression (Supplementary Table 2 and Fig. 6). Human samples with OA scores ranging from 18 to 20 had diffuse Safranin O staining (Fig. 6a), which was coupled with significantly higher number of β-catenin[+] cells (Fig. 6b) relative to human samples that had lower OA scores (14–16) and late-stage OA samples (24). These data point to the possibility that in diseased human TMJ fibrocartilage, canonical Wnt signals may be enhanced. Consequently, inhibition canonical Wnt signalling via Wnt inhibitors such as SOST could be beneficial to promote FCSCs to repair fibrocartilage and potentially prevent degenerative disease.

## Discussion

The cartilage of the mandibular condyle is considered fibrocartilage due to the presence of a fibrous, SZ tissue lining the condyle surface critical for providing tensile biomechanical properties[3]. Here, we define a novel function for this fibrous tissue as a specialized niche harbouring a reservoir of FCSCs through several lines of evidence. First, we show that the SZ tissue harbours slow-cycling LRCs and αSMA[+] cells[38] that differentiate into mature chondrocyte progeny. FCSCs isolated from the SZ tissue are clonogenic and multipotent *in vitro*. Most strikingly, a single FCSC can spontaneously differentiate into chondrocytes and osteocytes *in vivo* (Supplementary Fig. 10). Unlike tissue-specific stem/progenitor cells from other musculoskeletal tissues, such as tendon[42] or tooth[54], FCSCs generate bone with a haematopoietic microenvironment without the assistance of osteogenic reagents such as hydroxyapatite/tricalcium phosphate (HA/TCP) ceramic, Matrigel or BMP-2. Unlike BMSCs[33,55], FCSCs spontaneously form cartilage anlagen and can generate bone through both intramembranous and endochondral ossification. Thus we identify FCSCs that are inherently programmed to generate cartilage and bone tissues and spontaneously recapitulate bone developmental processes *in vivo*.

Premature stem/progenitor cell depletion is implicated in intervertebral disc degeneration[56] and other disease pathologies including craniosynostosis[57], cardiovascular disease[58] and lupus[59]. Our results reveal a novel paradigm for TMJ degenerative pathology, where the maintenance of FCSC pool in the SZ niche is critical for TMJ fibrocartilage homoeostasis and integrity. In healthy adult cartilage, chondrocytes exist under normal conditions of hypoxia and relatively low metabolic turnover[60]. During OA, enhanced biosynthesis and cellular loss are evidence of failed reparative mechanisms, whereby the chondrocyte's reparative capabilities are overridden by the tissue's functional demands[61]. Consistent with this idea, in two TMJ degeneration animal models[29,35] we show that increased proteoglycan deposition and OA histopathology[50,51] are also coupled with depletion of FCSCs localized within the SZ niche. This degenerative mechanism is in contrast to knee hyaline articular cartilage, where OA pathology has been attributed to the limited regenerative capabilities of migratory chondrogenic progenitor cells potentially derived from the bone marrow[17]. Our findings signify that TMJ fibrocartilage therapies could target preserving and harnessing the regenerative capabilities of endogenous FCSCs within the SZ niche.

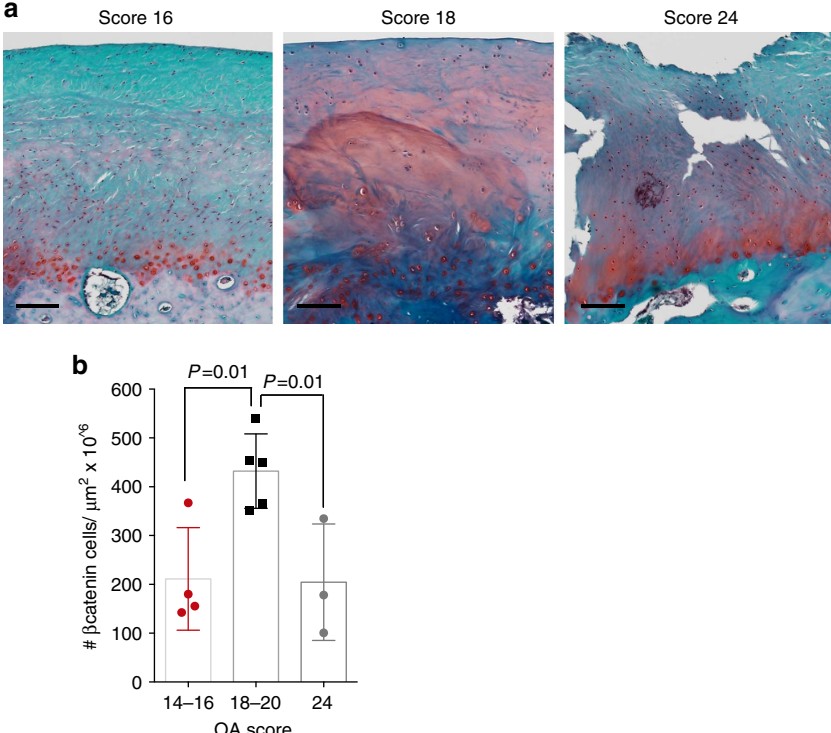

**Figure 6 | Aberrant Wnt signalling in osteoarthritic human TMJ fibrocartilage.** (**a**) Diseased human TMJ fibrocartilage samples excised from patients undergoing TMJ condyle replacement surgery were stained with Safranin O. Observers (*n* = 2) blindly assessed OARSI histopathological osteoarthritic scores. Scale bar, 125 μm. (**b**) β-catenin expression was evaluated by immunohistochemistry and the number of cells expressing β-catenin was quantified using cellSense Imaging Software. Data are mean ± s.d.; *n* = 3–5 patient samples; one-way ANOVA followed by Tukey's *post hoc* test. ANOVA, analysis of variance.

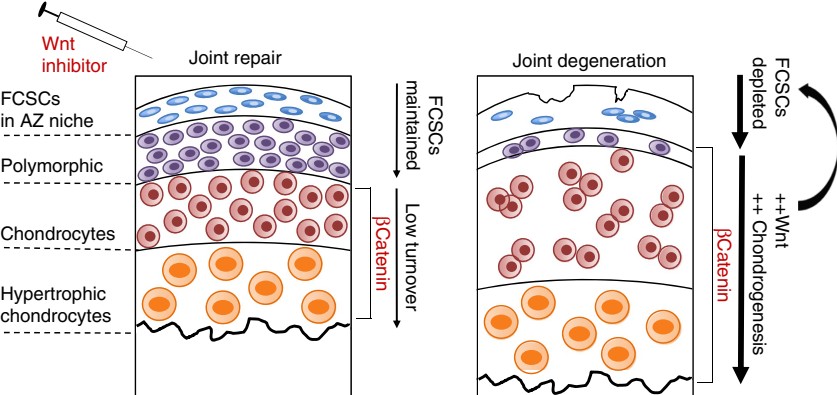

**Figure 7 | Therapeutic strategy to harness endogenous FCSCs to regenerate and repair cartilage.** Healthy adult TMJ fibrocartilage exists under low metabolic activity. A reservoir of FCSCs resides in SZ niche, canonical Wnt signalling is controlled and β-catenin is restricted to mature chondrocytes. During TMJ degeneration overactive Wnt signalling enhances chondrocyte maturation, depletes FCSC pool in SZ niche, and contributes to the pathological progression of degeneration. Therapeutic application of exogenous canonical Wnt inhibitor maintains FCSC pool and repairs cartilage.

The maintenance of stem/progenitor cell pool relies on intrinsic, local and systemic signalling cues. We have identified canonical Wnt as a signalling system critical for maintaining FCSC pool and fibrocartilage tissue homoeostasis (Fig. 7). A body of evidence supports a role for the fine-tuning of canonical Wnt in mediating cartilage homoeostasis at different stages of chondrogenesis; canonical Wnt inhibits early chondrocyte commitment and promotes terminal differentiation[26–28,44,45]. Our data supports a model where inhibited canonical Wnt signalling promotes FCSC differentiation into mature chondrocytes. We further show that FCSCs treated with canonical Wnt inhibitor sclerostin can generate cartilage that is sustained for a longer period in our transplantation model. Overactive Wnt signalling has been shown to disrupt cartilage homoeostasis and contribute to OA pathological progression in several mouse models[29,62,63]. Accordingly, we show that β-catenin expression is upregulated and proteoglycan content is abnormal in both animal and human diseased TMJ fibrocartilage. We provide evidence to support a model where overactive Wnt signals cause depletion of FCSCs and contribute to degeneration.

To further corroborate our findings, we show that treatment with exogenous canonical Wnt inhibitor SOST sustains FCSC pool and repairs cartilage using a rabbit TMJ injury model. While the experimental rabbits exhibited little change in dietary intake and overall body mass, the biomechanical properties and bone phenotype of SOST-treated condyles were not evaluated in these present studies. Before the application of this regenerative strategy in humans, future studies using pre-clinical animal models are required to include long-term cartilage and bone phenotypic recovery and extensive biomechanical analyses to confirm that functional joint recovery is achieved. Nonetheless, we show for the first time the promise of Wnt modulation in endogenous FCSCs as a non-invasive therapeutic strategy that could be used to treat fibrocartilage disease or injury. Small molecule Wnt inhibition has also been used therapeutically to treat myocardial infarction[64] and colon cancer[47]. The application of local small molecule Wnt inhibitors into fibrocartilage joint spaces offers a minimally invasive therapy following fibrocartilage injury. This strategy of injecting small molecules to manipulate endogenous FSCSs offers the advantages of bypassing invasive surgical interventions and minimizing the toxicity and side effects associated with the use of systemic drugs.

Although fibrocartilage tissues are inclusive of the TMJ condyle and disc, knee meniscus and intervertebral disc, the development, function, biomechanical properties and cellular and molecular phenotypes are quite different among each type of fibrocartilage tissue[1–4]. Consequently, it is likely that the stem/progenitor cell constituents making up these fibrocartilaginous tissues also vary in terms of behaviour, microenvironment niche and fate determination mechanisms. Thus, 'a one size fits all' therapeutic template for exploiting resident FCSCs may not apply and therapies should be adjusted and optimized for each fibrocartilage tissue type accordingly. For example, in the intervertebral disc the survival and maintenance of Tie2$^+$GD2$^+$ nucleus pulposus progenitor cells are dependent on angiopoietin-1 ligand[56]. Furthermore, in knee meniscus fibrocartilage, both CTGF and TGFβ3 are critical for meniscus regeneration[16,18]. These differences in fibrocartilage tissue properties and cells bring to light the necessity of extensive characterization and analyses of tissue-specific fibrocartilage stem/progenitor cells to facilitate the development of effective and tailored regenerative approaches.

Our present study focusing on exploiting FCSCs in the TMJ raises the possibility of applying this therapeutic strategy to hyaline articular cartilage in other joints, such as the knee and hip. However, unlike the TMJ, both knee and hip joints are load bearing and are not lined with fibrocartilaginous tissue[3,35]. Thus it is likely that endogenous stem/progenitor cells would have to be recruited from other sites to repair hyaline articular cartilage. In corroboration of this idea, studies have shown that migratory chondro-progenitor cells (CPCs) are recruited to osteoarthritic cartilage via capillary ingrowth through tidemark breaks into the cartilage; however CPCs are only found in late-stage osteoarthritic cartilage after significant cartilage destruction has occurred[17]. Alternatively, other studies have identified articular cartilage progenitor cells on the surface of knee articular cartilage[65], but their cartilage regenerative capabilities have yet to undergo rigorous testing for translational applications[66]. No studies to date have identified a population of stem cells in hyaline joint cartilage capable of spontaneously generating cartilage and bone with an organized HME when transplanted *in vivo*. Taking into consideration the difficulty of identifying effective resident stem/progenitor cells for hyaline articular cartilage repair, it may be advantageous to test the ability of FCSC transplantation to repair hyaline cartilage defects in knee and hip joints.

Taken together, we discover that resident stem cells in fibrocartilage localized within SZ niche in the TMJ condyle that possess potent chondrogenic and osteogenic potential. We also present a potential targeted therapeutic strategy for fibrocartilage degenerative disease, whereby regulation of canonical Wnt signals can sustain FCSC pool and maintain tissue

homoeostasis. These studies shed new light on the development of potential fibrocartilage therapies that exploit endogenous stem cells' regenerative capabilities.

## Methods

**Animals.** Male athymic nude mice ages 6–8 weeks, Sprague Dawley male rats ages P0 and 6–8 weeks, and male New Zealand White rabbits ages 12 weeks were used with approval from the Institution of Animal Care and Use Committee at Columbia University Medical Center (AC-AAAC2908; AC-AAAD1375; AC-AAAD9804; AC-AAAF4205). Eight-week-old male transgenic rat GFP[43] tissues, and $sost^{-/-}$ tissues[49] from 4- to 24-week-old male mice were provided by Dr Dongming Sun (Rutgers University) and Dr Tzong-Jen Sheu (Rochester University), respectively.

**Label-retaining cells.** EDU (Invitrogen, A10044) was injected into E16.5-E18.5 pregnant rat daily (Invitrogen, 50 mg g$^{-1}$ body mass). EDU labelled cells were detected in offspring on paraffin embedded sections (Invitrogen C-10420) and quantified (ImageJ).

**Histology and histomorphometry.** Tissue samples were fixed in 4% paraformaldehyde, decalcified in EDTA, and prepared for either paraffin or frozen embedded sections. Serial tissue sections were stained haematoxylin and eosin, Safranin O, toluidine blue or tartrate-resistant acid phosphatase (TRAP; Sigma 387A). For immunohistochemistry serial sections were sections were enzymatically treated with Chondroitinase ABC (Fisher C3667-10UN) and immunolabelled with primary antibodies at 4 °C overnight followed by secondary antibody to detect immunoactivity (Supplementary Table 3). Isotype-matched negative control antibodies were used under the same conditions. The area of antibody expression or chemical stain and the cell number were quantified using Olympus cellSens Dimension imaging software.

**Lineage tracing.** The generation of transgenic αSMACreERT2 mice was previously reported[38]. The αSMACreERT2 dual transgenic mice were generated by breeding αSMACreERT2 male mice with Ai9 female mice (Jackson Labs, #007905). The Ai9 mice harbour targeted mutation of Gt(ROSA)26Sor locus with a loxP- flanked STOP cassette, preventing transcription of a CAG promoter-driven red fluorescent protein variant (tdTomato). For *in vivo* lineage tracing studies, 16 day-old αSMACreERT2/Ai9 mice were treated with 62.5 mg g$^{-1}$ tamoxifen per mass and killed after 2 and 15 days. The αSMACreERT2/Ai9-positive untreated mice and αSMACreERT2/Ai9-negative treated mice were evaluated as negative controls.

**Cell isolation and culture.** TMJ discs and condyles were dissected from Sprague Dawley rats ages 6–8 weeks. To isolate SZ tissue, whole TMJ condyles were digested with 4 mg ml$^{-1}$ dispase II (Roche 04942078001) for 15 min at 37 °C and SZ tissue was separated from the condyle using a dissecting microscope. Primary cells from condyles and SZ tissues were isolated[25]. Tissues were digested in dispase II/collagenase I (4 and 3 mg ml$^{-1}$). Donor-matched BMSCs were isolated from long bone tibia[67]. GFP TMJ cells were isolated from GFP rat[43] (tissues kindly donated by W. Young & D. Sun, Rutgers University, New Jersey). Single-cell suspensions were cultured (5% $CO_2$, 37 °C) in basal medium consisting of DMEM (Invitrogen 11885-092) supplemented with 20% lot-selected fetal bovine serum (FBS Hyclone), glutamax (Invitrogen 35050-061), penicillin-streptomycin (Invitrogen 15140-163) and 55 μM 2-mercaptoethanol (Gibco) for 4–6 days. Cells were detached with trypsin-EDTA (GIBCO) and plated at P1 for the *in vitro* experiments.

*RNA isolation and qRT-PCR.* Total RNA was purified from whole TMJ tissues (Qiagen 74704) or cells (Ambion 12183018A) and treated with DNAse I (Ambion AM2222) to remove genomic DNA. RNA quantity and purity was determined using Nanodrop. RNA samples (260/280 ≥ 1.8) were used to obtain cDNA (Biorad AM2222). Quantitative RT-PCR was performed using TaqMan Universal PCR Master Mix (Applied Biosystems 4304437) and pre-designed primers (Supplementary Table 4). Gene expression levels were normalized to housekeeping gene *gapdh* (Applied Biosystems Rn01775763_g1*).

**FACS analysis.** All fluorescence-activated cell sorting analysis was performed at Columbia Center for Translational Immunology Flow Cytometry Core (CCTI, Columbia University Medical Center, New York, NY). For flow cytometric analysis, FCSCs and BMSCs were immunolabeled with 1 μg of fluorescent conjugated antibodies or isotype-matched IgG controls (Supplementary Table 4) for 20 min at room temperature. Samples were analysed using BD FACSCanto II and calculated using FloJo program. Gating strategy for selecting positive cells involved a positive compensation control containing a single fluorochrome antibody and negative gating control including all but one fluorochrome antibody (Supplementary 1b). BD Influx cell sorter was used to isolate single GPP$^+$ TMJ-PC into 96-well plates.

**Multi-lineage differentiation.** Multi-lineage differentiation was tested *in vitro* using chemically defined media for chondrogenesis, osteogenesis, and apdipogenesis[25,42]. For chondrogenesis, cells ($1 \times 10^6$ per pellet) were pelleted in 15 ml polypropylene tubes by centrifugation and cultured (5% $CO_2$, 37 °C) for 3 weeks in Dulbecco's Modified Eagle medium supplemented with $10^{-7}$ mol l$^{-1}$ dexmethasone, 100 μmol l$^{-1}$ ascorbic acid, 1% insulin, transferrin, selenium (ITS), 1 mmol l$^{-1}$ pyruvate, and 10 ng ml$^{-1}$ TGF-β1. To induce osteogenesis, cells ($5 \times 10^4$) were culture in 12-well plate for 4–5 weeks in media containing αMEM supplemented with 20% FBS, dexamethasone ($10^{-8}$ M), 100 μM L-ascorbic acid, and 2 mM β-glycerophosphate. To induce adipogenesis, cells ($5 \times 10^4$) were cultured in 12-well plate using commercial adipogenic media for 1 week (Gibco A1007001). Calcium nodules and fat were visualized by staining with alizarin red and Oil Red O, respectively.

*In vivo transplantation.* Primary cells or single-cell colonies ($2.0 \times 10^6$) were seeded onto either collagen sponge (12 × 12 × 3 mm; Helistat 1690ZZ), gel foam (12 × 12 × 2 mm, Pfizer) or BD Matrigel (200 μl, BD Biosciences 354230)[68] and transplanted onto the dorsum of athymic nude mice (Harlen Athymic Nude-Foxn1$^{nu}$). Transplants were harvested after 2, 3, 4, 6 and 8 weeks for histologcal analysis.

*Transient transfection and luciferase activity.* Transient transfection of FCSCs was performed using the Neon Transfection System and kit (Invitrogen MPK1025). Optimal transfection conditions were determined using a GFP DNA plasmid to achieve approximately 60–80% transfection efficiency for approximately 72 h. Luciferase reporter constructs TopFlash and S33y plasmid (1 μg) or control vector and LacZ plasmid were transfected into FCSCs ($5 \times 10^5$ cells). LacZ plasmid was used as an internal transfection control. The transfected cells were split into three wells of 96-well plates and culture for 24 h. Luciferase activity was measured after 72 h (ref. 68).

**TMJ injury model.** TMJ degeneration was surgically induced in New Zealand white rabbits[35]. A punch biopsy was used to create a 2.5 mm perforation in the TMJ disc bilaterally. A periosteal elevator was placed under the TMJ disc to prevent damage to the condyle during disc perforation. One week post surgery, 0.1 ml SOST (100 ng ml$^{-1}$ in PBS, R&D 1406-ST-025) was injected into the inferior TMJ intra-articular space unilaterally once weekly. Vehicle control (PBS) was injected into the contralateral TMJ.

**Human analyses.** All human TMJ condyle samples were derived from subjects recruited by the National Institute of Dental and Craniofacial Research's Temporomandibular Joint Implant Registry and Repository (NIDCR's TIRR) in accordance with University of Minnesota IRB Human Subjects Code Number 0210M33782. Subjects who underwent TMJ surgery with and without the placement or removal of a TMJ implant were invited, by their doctor/surgeon, TMJ Clinician or NIDCR's TIRR Recruiter/Study Coordinator to participate, then consented and enrolled in NIDCR's TIRR. TMJ specimens were de-identified and prepared in paraffin blocks and banked at NIDCR sponsored TIRR[52] housed at the University of Minnesota. All samples were histologically scored[53] by two independent observers.

**Statistical analysis.** All statistics were calculated using Prism 6 Graphpad Software. The statistical significance between two groups was determined using paired Student's *t*-test assuming Gaussian distribution. The normality of distribution was confirmed using Kolmogorov–Smirnov test and the resulting two-tailed P value ≤ 0.05 was regarded as statistically significant difference. Among three groups, one-way analysis of variance followed by Tukey's *post hoc* test was used for statistical comparisons. For multiple comparisons, a two-way analysis of variance followed by Tukey's *post hoc* was used for statistical comparisons.

**Data availability.** The data that support the findings of this study are available from the corresponding author upon request.

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

## Acknowledgements

This investigation was supported by NIH grants K99DE022060-01A (to M.C.E.), 5R00DE0220660 (to M.C.E.), and R01DE021134 (to H.Y.) and Columbia University College of Dental Medicine. FACS was performed in the CCTI Flow Cytometry Core with the assistance of Dr Siu-Hong Ho, supported by NIH grants S10RR027050 and S10OD020056. Dr Sandra Myers co-director of the TMJ Implant Registry and Repository at the University of Minnesota, supported by NIH grant NO1-DE-22635 provided human TMJ samples. At Columbia University, we thank Dr Alex Romanov and the ICM staff for assistance with rabbit surgeries.

## Author contributions

M.C.E. conceived study, designed/performed experiments, analysed the data, prepared figures and wrote manuscript. M.C. designed study and experiments. S.P. performed immunohistochemistry and histomorphometry. D.K. and G.M.I. performed qRT-PCR and immunohistochemistry. I.K. performed lineage-tracing experiment. H.Y. and C.S. performed μCT analysis. D.S. and T.-J.S. provided GFP and *sost* mutant tissues, respectively. D.A.K. and A.K. performed rabbit surgeries. J.J.M. provided oversight.

## Additional information

**Competing financial interests:** The authors declare no competing financial interests.

