## [Peer Review File · Nature Communications]

Reviewers' comments:

Reviewer #1

Expert in TMJ cartilage regen
(Remarks to the Author):

Overall Review

This is a valuable manuscript for the TMJ field in terms of understanding resident cell populations for regeneration. The strengths of the manuscript were the extensive work including multiple animal models and human tissue, and the controlled design of experiments. Minor concerns are listed below.

The primary concern was that throughout the manuscript there are exaggerated claims based on results. Claims can be toned down. For example, Page 5: "Cartilage markers (sox6, sox9, acan, col2a1 and col10a1) were significantly decreased in AZ compared to CC (Fig. 1G), suggesting that AZ may harbor stem cells." How does this suggest the presence of stem cells?

Minor concerns:

- 1) "TMJ condyle articular zone (AZ) and mature condylar cartilage zones (CC)" - can the anatomy be more clearly defined? In the methods, this was a major concern that the true meaning of and delineation between the AZ and CC were not well explained. Can the methods be more clear about how they were separated? Moreover, can these be better described in terms of more conventional notation such as superficial zone, proliferative zone, and middle/deep zone, or equivalent?
- 2) Page 5: "AZ LRCs were significantly higher than CC LRCs". This does not make sense, please rephrase. This is a common concern throughout the manuscript. For example, saying that cells decreased in the following sentence. I.e., a cell does not decrease.
- 3) Page 5: "ECM markers type I collagen (col1a1) and lubricin (prg4) were upregulated in AZ". Relative to what? From Fig. 1F, it can be seen that it's relative to CC, but the preceding sentence says that only the AZ was surgically separated from the condyles. Requires language clarification.
- 4) The manuscript in general would benefit from grammar proofreading.
- 5) Page 10: "The TMJ condyle is considered fibrocartilage." Should be "The cartilage of the mandibular condyle is considered...", as the condyle is predominantly bone.
- 6) Page 11, incomplete sentence: "First, articular zone harbor LRCs and α SMA+ cells⁴¹ that give rise to mature chondrocyte progeny."
- 7) Table: Use two significant figures for standard deviation, and ensure that the decimal place of the mean matches that of the standard deviation.
- 8) Figure 3D: scale bar length missing

Reviewer #2

Expert in fibrocartilage repair
(Remarks to the Author):

This study addresses an important clinical issue, which is healing and regeneration of fibrocartilaginous containing tissues. In this specific study they examine fibrocartilage in the temporo-mandibular joint, but the findings could ultimately be applicable to fibrocartilage in the intervertebral disc as well as knee joint meniscus. They find a pool of endogenous progenitor cells in the superficial zone of fibrocartilage in the TMJ that appears to play a role in maintenance of the tissue, and to participate in repair.

The general approach of trying to identify and ultimately stimulate intrinsic stem cells as opposed to exogenous cell transplantation certainly has great potential. The authors present a series of experiments which demonstrates the existence of fibrocartilage stem/ progenitor cells in the superficial region of the temporomandibular joint. They go on to show that the cells are controlled by and responsive to Wnt signalling. Specifically, inhibition of Wnt signaling (with sclerostin) promotes maintenance of a fibrocartilage phenotype and leads to fibrocartilage regeneration in their in vivo work.

The authors present rigorous data evaluating the structure and composition of tissue at the temporomandibular joint. However, a weakness is that there are no functional assays that evaluate the biomechanical properties of this tissue. This could be mentioned as a limitation in the discussion section.

I also do not fully understand the α -SMA part - please clarify how this identifies the FCSC pool.

In considering the clinical aspects, injury and pathology of other fibrocartilage containing tissues is much more common, including intervertebral disc in the spine and meniscus in the knee. TMJ pathology is much less common. The authors briefly mention other tissues, implying that a similar stem cell pool may be present in disc and meniscus. However, these are all very different tissues, in very different microenvironments, and it may be a leap to assume similar mechanisms. Given the importance of intervertebral disc and meniscus injury, the authors might add further discussion about how the current findings might apply to those tissues.

Reviewer #3

Expert in regen and stem cells

(Remarks to the Author):

The purpose of this study was to demonstrate that fibrocartilage stem cells (FCSCs) exist within the articular zone of the temporomandibular joint (TMJ) and to determine the signaling pathways that maintain their ability to regenerate both osseous and cartilaginous structures. The underlying goal of the study was to identify a biological target for articular cartilage regeneration by harnessing the intrinsic capabilities of resident FCSCs. The authors showed that FCSCs exist in a fibrocartilaginous niche within the articular zone of the TMJ and that a single FCSC within this niche can regenerate bone and cartilage in addition to establishing the framework for the initiation of hematopoiesis. They also showed that inhibition of the canonical Wnt pathway sustains the pool of FCSCs available to regenerate articular cartilage. Overall, this study is well-written; the study rationale, methods, results, and corresponding figures are presented clearly.

The following is a list of critiques and suggestions that I believe may help improve the impact of this manuscript.

- 1) Although it is presumed that the target audience includes those within the field of oral and/or maxillofacial surgery, the overall impact of the paper could be improved by including a brief discussion of the potential applications in other clinical specialties that involve the treatment of articular cartilage problems, including orthopaedic surgery and rheumatology. For example, is the articular surface of the TMJ similar in structure to those of other joints, such as the knee, hip, and shoulder?
- 2) Why not test the use of the FCSCs in osteochondral defects using an animal model? It would be important to determine whether the cells need BMPs to promote AC repair. This would increase the impact of the work.
- 3) The authors show that a single FCSC can produce cartilage and bone mimicking endochondral

ossification. However, for the purpose of cartilage repair, it would be important to demonstrate that the cells remain in the cartilage stage and do not become bone. It would also be important to determine whether these cells are sensitive to angiogenesis since angiogenesis is an important feature that promotes the transition from cartilage into bone.

4) I think the authors should make clear that their animal model most closely resembles the pathophysiology of secondary (post-traumatic) osteoarthritis (OA) than that of primary (idiopathic) OA.

5) It would be advantageous to include any results related to subchondral bone, if possible, because it is the alteration of stress distribution transmitted through the subchondral bone that leads to bony overgrowth, transition of cartilage into bone, joint malformation, and rapid chondral wearing.

6) The authors should mention that their study was approved by their local IACUC and/or IRB, if applicable.

Overall, this study is well-written; the study rationale, methods, results, and corresponding figures are presented clearly. As described above, trying to make a parallel with articular cartilage repair would be a tremendous improvement in the paper. In other words, how does the articular cartilage of the TMJ differ from that of other joints, such as the knee, hip, and shoulder? How can these findings regarding FCSCs be translated into cell therapies for conditions related to articular cartilage? Demonstrating that FCSCs can repair articular cartilage in other joints (instead of making cartilage under the skin) would be very important.

Thank you for the opportunity to review this manuscript.

Detailed Response to the Reviewers

We thank all the reviewers and have carefully considered all their comments. We have addressed their comments point by point below:

REVIEWER 1

Comment 1:

The primary concern was that throughout the manuscript there are exaggerated claims based on results. Claims can be toned down. For example, Page 5: "Cartilage markers (sox6, sox9, acan, col2a1 and col10a1) were significantly decreased in AZ compared to CC (Fig. 1G), suggesting that AZ may harbor stem cells." How does this suggest the presence of stem cells?

Response:

We thank the reviewer for his/her insightful comment and regret our exaggerated claims based on the results. Thus we have revised the text within the entire results section to tone

down our scientific interpretations and conclusions. For example we have rewritten this particular sentence "SZ tissue may harbor undifferentiated cells, while CC tissue may harbor mature cell phenotypes, including chondrocytes." Please **see Results section**.

Comment 2: "TMJ condyle articular zone (AZ) and mature condylar cartilage zones (CC)" - can the anatomy be more clearly defined? In the methods, this was a major concern that the true meaning of and delineation between the AZ and CC were not well explained. Can the methods be more clear about how they were separated? Moreover, can these be better described in terms of more conventional notation such as superficial zone, proliferative zone, and middle/deep zone, or equivalent?

Response:

We regret the lack of clarity in the cartilage notation in the original manuscript. We have now changed the notation of articular zone (AZ) to more conventional notation of superficial zone (SZ) within the body of the manuscript. During development the TMJ condyle is a site for endochondral ossification in the mandible, whereby the zones of maturation can be divided into superficial, polymorphic/proliferative, chondrocyte and hypertrophic chondrocyte zones of maturation or stages of differentiation. In adults, after mandibular growth is completed, the TMJ condyle can be described in more conventional articular cartilage notation including, superficial, polymorphic, middle, and deep cartilage zones. The superficial zone is by in large the fibrous portion of the condyle containing more undifferentiated cell phenotypes. Regardless of the developmental stage, the superficial zone is distinct from the other cell zones, given it is comprised of fibrous tissue. We use the term condylar cartilage (CC) to include the remaining mature, cartilaginous zones: polymorphic/proliferative, chondrocyte, and hypertrophic chondrocyte. We have now described the anatomy and terminology describing these zones of maturation during development and in adults in the introduction (**see Introduction page 3**). We further describe which cells are included in the notation AZ and CC in the results section (**see Results page 5**).

Comment 3: Page 5: "AZ LRCs were significantly higher than CC LRCs". This does not make sense, please rephrase. This is a common concern throughout the manuscript. For example, saying that cells decreased in the following sentence. I.e., a cell does not decrease.

Response: We regret our grammatical error that does not identify or state what changes precisely throughout the original manuscript. We have now corrected these errors within the **Results** section in the revised manuscript. For example we write "the percentage of SZ LRCs was significantly higher than the percentage of CC LRCs." Another example is we also rewrote "the percentage of α SMA⁺col2a1⁻ cells in SZ was significantly higher than the percentage of α SMA⁺col2a1⁺ cells in CC.

Comment 4: Page 5: "ECM markers type I collagen (col1a1) and lubricin (prg4) were upregulated in AZ". Relative to what? From Fig. 1F, it can be seen that it's relative to CC, but

the preceding sentence says that only the AZ was surgically separated from the condyles. Requires language clarification.

Response: We regret our language that does not specify which groups were being compared throughout the original manuscript. We have now carefully reviewed the manuscript to ensure we changed the language so that we clearly state which groups were being compared. For example, we write “ECM markers type I collagen (*col1a1*) and lubricin (*prg4*) were upregulated in SZ relative to CC tissue.”

Comment 5: The manuscript in general would benefit from grammar proofreading.

Response: We regret our oversight in grammatical errors. We have revised the original manuscript and have had multiple scientists review, proofread and revise the manuscript for grammatical errors.

Comment 6: Page 10: "The TMJ condyle is considered fibrocartilage." Should be "The cartilage of the mandibular condyle is considered...", as the condyle is predominantly bone.

Response: We have made this correction in the revised manuscript (see **Page 11**).

Comment 7: Page 11, incomplete sentence: "First, articular zone harbor LRCs and α SMA+ cells⁴¹ that give rise to mature chondrocyte progeny."

Response: In the revised manuscript we have now rewritten the sentence to be a complete sentence. We write: “First, we show that superficial zone harbor LRCs and α SMA+ cells⁴¹ that differentiate into mature chondrocyte progeny.” (see **Page 12**)

Comment 8: Table: Use two significant figures for standard deviation, and ensure that the decimal place of the mean matches that of the standard deviation.

Response: For ease of readability and better visualization we have now reformatted Table 1 into a bar graph in Supplementary Figure 1. Our intent in Figure 1 is to demonstrate that BMSCs and FCSCs share similar expression pattern of cell surface markers.

Comment 9: Figure 3D: scale bar length missing

Response: We have now added a scale bar length to Figure 3D in the revised manuscript.

REVIEWER 2

Comment 1: The authors present rigorous data evaluating the structure and composition of tissue at the temporomandibular joint. However, a weakness is that there are no functional assays that evaluate the biomechanical properties of this tissue. This could be mentioned as a limitation in the discussion section.

Response: We thank the reviewer for his/her insightful comment regarding a limitation on these studies. We point out this limitation in the body of the discussion and also note that functional recovery of the joint needs to be addressed using this approach in future studies (see **Discussion Pages 13-14**).

Comment 2: I also do not fully understand the alpha-SMA part - please clarify how this identifies the FCSC pool.

Response: We performed a lineage tracing study and used alpha-SMA^{CreERT2}/Ai9 mice to trace the origin of mature chondrocytes in the TMJ condyle. A cell lineage is the developmental origin of a cell traced back to the cell from which it arises. In this experiment our goal was to demonstrate that the undifferentiated fibrocartilage stem cells localized in superficial zone niche differentiates and gives rise to the mature cell phenotypes including chondrocytes localized within the proliferative, chondrocyte and hypertrophic chondrocyte zones. In a separate study performed by a co-author on this manuscript Ivo Kalajzic (*Stem Cells*, 2012), alpha-SMA was shown to be a marker for fibrocartilage and skeletal progenitors. In alpha-SMA^{CreERT2}/Ai9 the expression of the Cre molecule is activated under the control of the α SMA promoter upon treatment with tamoxifen. Thus, α SMA⁺ cells can be tracked during development by activating the expression of the tdTomato visual reporter through Cre-mediated recombination upon tamoxifen treatment. In our experiments (Figs 1H-J), we show that α SMA⁺ cells were initially localized in the superficial zone, did not express mature chondrocyte marker type II collagen, and thus may represent FCSCs. However, following growth after 2 weeks α SMA⁺ cells from the superficial zone differentiated into mature chondrocytes and expressed type II collagen. Thus our goal was to show that α SMA⁺ cells residing in the superficial zone may represent a pool of undifferentiated, FCSCs. During growth and development, these α SMA⁺ cells potentially representing FCSCs differentiated into mature chondrocytes localized within the proliferative, chondrocyte

and hypertrophic chondrocyte zones. We regret that this intent of this experiment was not explicit in the original manuscript. We have now clarified this lineage tracing experiment within the revised manuscript (see **Page 5, last paragraph**). We added this sentence: “We hypothesized that SZ cells may give rise to mature chondrocytes localized within CC tissue. To determine the origin of mature chondrocytes in the TMJ condyle, we performed a lineage tracing study and the skeletal stem/progenitor cell marker α SMA was traced in α SMACreERT2/Ai9 transgenic mice”.

Comment 3: In considering the clinical aspects, injury and pathology of other fibrocartilage containing tissues is much more common, including intervertebral disc in the spine and meniscus in the knee. TMJ pathology is much less common. The authors briefly mention hear other tissues, implying that a similar stem cell pool may be present in disc and meniscus. However, these are all very different tissues, in very different microenvironments, and it may be a leap to assume similar mechanisms. Given the importance of intervertebral disk and meniscus injury, the authors might add further discussion about how the current findings might apply to those tissues.

Response: We thank the reviewer for this insightful comment. We agree that while we extensively characterized stem cells within the TMJ condyle fibrocartilage, it is unlikely that an identical population of stem cells is present within other fibrocartilage tissues. Indeed fibrocartilages including the TMJ, knee meniscus and IVD have disparate anatomical functions and most likely have different cell constituents with varying microenvironments. Thus, a universal therapeutic approach to harness fibrocartilage stem cells would have to be tailored for each fibrocartilage type. We have presented just one example of a therapeutic approach to harness fibrocartilage stem cells in TMJ condyle tissues. Harnessing stem cells in knee meniscus and intervertebral disc would have to be tailored to tissue-specific stem cell behavior/function and micro-environment. We have expanded upon this topic within the discussion of the revised manuscript (see **Discussion, fifth paragraph**).

REVIEWER 3

Comment 1: Although it is presumed that the target audience includes those within the field of oral and/or maxillofacial surgery, the overall impact of the paper could be improved by including a brief discussion of the potential applications in other clinical specialties that involve the treatment of articular cartilage problems, including orthopaedic surgery and rheumatology. For example, is the articular surface of the TMJ similar in structure to those of other joints, such as the knee, hip, and shoulder?

Response: We agree that other clinical specialties related to joints, particularly orthopaedics, would be interested in our present study. The TMJ articular surface is fibrocartilage and is dissimilar to other joints comprised of hyaline cartilage, such as the knee

and hip. Therefore, it is unlikely that an identical population of stem cells found in the TMJ condyle fibrocartilage would be found in the knee or hip. Other groups have reported discovery of progenitor cell populations in knee hyaline cartilage (Koelling et al. 2009) (Dowthwaite et al. 2004), with regenerative capabilities that differ from FCSCs identified in our present study. We have now included new references and a paragraph highlighting this topic in the revised manuscript (see **Discussion, sixth paragraph**).

Comment 2: Why not test the use of the FCSCs in osteochondral defects using an animal model? It would be important to determine whether the cells need BMPs to promote AC repair. This would increase the impact of the work.

Response: We thank the reviewer for his/her excellent comment. We also agree that the ability of transplanted FCSCs to engraft and repair an osteochondral defect would be a powerful example of the application of FCSC transplantation for cartilage repair and regeneration. However, stem cell transplantation approaches to repair cartilage and other musculoskeletal tissues have faced many clinical obstacles such as immune rejection and pathogen transmission. Thus, the focus of the present study was to bypass the clinical translational barriers related to stem cell transplantation by harnessing the regenerative capabilities of resident stem cells instead (see **Introduction, second paragraph**). Therefore, testing the use of FCSCs in an osteochondral defect is out of the scope of the current study. Nonetheless, we strongly agree that it would be interesting and highly impactful to test the application of FCSCs to repair cartilage defects in the TMJ and other joints more commonly affected by osteoarthritis, including the knee or hip. In a separate study, we have future plans to test FCSCs engraftment into diseased TMJs, knee joints and/or knee osteochondral defects. Given that FCSCs spontaneously form cartilage when transplanted subcutaneously in the absence of exogenous BMPs, we surmise FCSCs would not require BMPs to promote AC repair. In the revised manuscript, we have now included a discussion regarding the possible approach of transplanting FCSCs to treat knee and hip cartilage degeneration (see **Discussion, sixth paragraph**).

Comment 3: The authors show that a single FCSC can produce cartilage and bone mimicking endochondral ossification. However, for the purpose of cartilage repair, it would be important to demonstrate that the cells remain in the cartilage stage and do not become bone. It would also be important to determine whether these cells are sensitive to angiogenesis since angiogenesis is an important feature that promotes the transition from cartilage into bone.

Response: Reviewer 3 makes several critical points in Comment 3. Relative to bone regeneration, cartilage regeneration and repair is more difficult, given the lack of vascular supply and limited number of cells present in cartilage. In our transplantation model, we show that FCSCs spontaneously form cartilage subcutaneously. However, we also show that the cartilage formed is transient and eventually remodels into bone after 8 weeks (see

Figure 3 and Supplementary Figures 2-4). To harness FCSC cartilage regenerative capabilities, the key for us was to identify cellular pathways that we could manipulate experimentally to help control and maintain the cartilage formed by FCSCs and also inhibit bone formation. We were prompted to investigate the Wnt pathway. A body of evidence supports that inhibition of Wnt signals promotes skeletal progenitor cells to differentiate toward cartilage lineage, while preventing skeletal progenitor cells commitment toward bone lineage. Furthermore, Wnt signals also promote osteoblast maturation and promote terminal differentiation of chondrocytes to undergo hypertrophy. We found a similar paradigm to be true for FCSCs (see **Supplementary Figure 9**). A single FCSC can spontaneously commit to both bone and cartilage lineages. Inhibition of canonical Wnt signals induces FCSC to differentiate into chondrocytes, while Wnt signals promote chondrocyte terminal differentiation and FCSC bone formation. We further show that when FCSCs are pre-treated with Wnt inhibitor sclerostin prior to transplantation subcutaneously, FCSCs have a sustained cartilage phenotype and also have a delayed/reduced bone formation (see **Figure 4F-G**). While our focus in this study was to manipulate/inhibit Wnt signaling to promote FCSC cartilage repair and regeneration, we acknowledge that the role of Wnt in FCSC bone repair and regeneration is another scientific direction that warrants further analyses.

Secondly, we also agree that a critical factor regulating cartilage remodeling into bone is new blood vessel formation. We currently have a separate study investigating this very topic. [Redacted] Our submission of this separate study is in the pipeline and pending publication of the current manuscript in review for *Nature Communications*.

Comment 4: I think the authors should make clear that their animal model most closely resembles the pathophysiology of secondary (post-traumatic) osteoarthritis (OA) than that of primary (idiopathic) OA.

Response: We agree that our OA animal model develops OA secondary to disc perforation injury. In the revised manuscript we have now clarified and added a sentence that our rabbit TMJ injury model developed OA secondary to disc perforation trauma (see **Results, page 10**).

Comment 5: It would be advantageous to include any results related to subchondral bone, if possible, because it is the alteration of stress distribution transmitted through the subchondral bone that leads to bony overgrowth, transition of cartilage into bone, joint malformation, and rapid chondral wearing.

Response: We have previously published that the rabbit TMJ injury model used in this study does not develop overt histopathological evidence of subchondral bone pathology until 12 weeks following disc perforation injury (*Osteoarthritis Cartilage*, Embree et al. 2015), which include osteophytes, rapid transition of bone, and heterotopic disc ossification. In the current study we only analyzed rabbits 8 weeks after introducing disc perforation injury and treatment with sclerostin. Thus we did not observe evident histopathological evidence of

bony changes. We suspect that if we analyzed rabbit TMJs 12 weeks following disc perforation, bony changes would be more evident. Nonetheless, our future plans are to test small molecules/potential candidate drugs that inhibit Wnt signaling in larger, pre-clinical models, such as mini-pig. For these studies, we anticipate developing single-dose or control release small molecules for modifying FCSCs and analyzing total TMJ tissues over a substantial time course. We anticipate our analyses will include bone characterization and also biomechanical and functional recovery as suggested by **Reviewer 2, Comment 1**). We have now acknowledged this limitation in our revised manuscript (see **Discussion, fourth paragraph**).

Comment 6: The authors should mention that their study was approved by their local IACUC and/or IRB, if applicable.

Response:

We have included the approved IACUC protocol numbers (AC-AAAC2908; AC-AAAD1375; AC-AAAD9804; AC-AAAF4205) in the **Methods** section.

Comment 6: Overall, this study is well-written; the study rationale, methods, results, and corresponding figures are presented clearly. As described above, trying to make a parallel with articular cartilage repair would be a tremendous improvement in the paper. In other words, how does the articular cartilage of the TMJ differ from that of other joints, such as the knee, hip, and shoulder? How can these findings regarding FCSCs be translated into cell therapies for conditions related to articular cartilage? Demonstrating that FCSCs can repair articular cartilage in other joints (instead of making cartilage under the skin) would be very important.

Response: We also agree that the identification of FCSCs that spontaneously regenerate cartilage under the skin leads to questions of whether FCSCs can be transplanted to regenerate and repair cartilage in other joints. Given that the focus of this study was to demonstrate that resident FCSCs can be stimulated to repair/regenerate cartilage in the absence of stem cell transplantation, experiments related to FCSC transplantation is out of the scope of this study. Nonetheless, this is an important and highly impactful scientific question that we agree warrants further investigation. We have current experiments underway to test the efficacy of FCSCs to engraft and repair not only cartilage, but also bone defects. Please also see our response to **Reviewer 3, Comment 2**. We have now added this point to the Discussion (see **Discussion, sixth paragraph**).

REVIEWERS' COMMENTS:

Reviewer #3 (Remarks to the Author):

The authors have answered all my primary concerns regarding the first review of the paper. Although I would have preferred to have them repair osteochondral defects using this new cell population and to clarify the role of angiogenesis in the process, I agree with the authors that these additional experiments are beyond the scope of the research paper.